# Beneficial Effects of Fermented Camel and Cow's Milk in Lipid Profile, Liver, and Renal Function in Hypercholesterolemic Rats

Yousef Mesfer Alharbi [1], Khaled Meghawry El-Zahar [2,3] and Hassan Mirghani Mousa [2,*]

1   Department of Veterinary Medicine, College of Agriculture and Veterinary Medicine, Qassim University, Buraidah 51452, Saudi Arabia; yrhby@qu.edu.sa
2   Department of Food Science and Human Nutrition, College of Agriculture and Veterinary Medicine, Qassim University, Buraidah 51452, Saudi Arabia; k.abdelsayed@qu.edu.sa
3   Food Science Department, Faculty of Agriculture, Zagazig University, Zagazig 44511, Egypt
*   Correspondence: hasmousa@hotmail.com; Tel.: +966-16380-16225; Fax: +966-16380-1360

**Abstract:** As hyperlipidemia has been associated with cardiovascular diseases, this study investigated the influence of probiotic-fermented camel and cow's milk on blood lipid profiles in hypercholesterolemic rats. When tested, probiotic-fermented camel and cow's milk exhibited the highest overall acceptance score in flavor and texture. Forty-eight male Wistar rats were divided into eight groups ($n = 6$). The first group served as normal control, while groups 2–8 were fed on a high-fat (HFD), high-cholesterol diet throughout the experimental period and treated with different types of fermented milks. Feeding rats on probiotic-fermented milk resulted in a significant decrease in the level of triglycerides (TG), cholesterol, and LDL compared with the positive control group. Albumin and total protein concentrations increased significantly, while ALT, AST, and creatinine were significantly reduced in rats fed on probiotic-fermented milk. The results indicated that probiotic-fermented milk might improve liver and kidney functions in hypercholesterolemic rats. These findings highlighted the ameliorative potentials of camel milk against hyperlipidemia and oxidative stress in rats.

**Keywords:** camel milk; probiotic bacteria; lipids profile; biochemical markers; fermentation; hypercholesterolemia



## 1. Introduction

Camel milk is the major accepted milk in the Arab Gulf nations. Camel milk is considered nutritionally higher than cow's milk due to the presence of protected proteins, such as lysozyme, which is an immunoglobulin, anti-inflammatory, and antioxidant. Camel milk contains no β-lactoglobulin, which may cause an allergic effect in some people. However, camel milk contains a high amount of iron, potassium, and vitamins C, E, and A [1]. Camel milk causes a significant decrease in insulin measurements in type 1 diabetes subjects, helping to obtain glycemic control along with significant improvement in HbA1c value and decrease in microalbuminuria [2,3]. Significant changes were also observed in body mass index, lipid profile, liver enzymes, albumin, urea, and creatinine in camel milk-fed laboratory subjects with insulin-dependent diabetes. Thus, dromedary camel's milk can carefully be suggested as an alternative drug to handle lipid anomalies and hypercholesterolemia correlated with diabetes [1,4]. Fermented camel milk containing *Bifidobacterium lactis* has been described to maintain a hypercholesterolemic effect in rats and in lowering of plasma and liver cholesterol concentrations [5,6]. The hypocholesterolemic mechanism of camel milk is still vague, but various designs have been suggested, including the communication between bioactive peptides from camel milk proteins and cholesterol, which results in cholesterol reduction, and the presence of orotic acid in camel milk, which is thought to be responsible for reducing cholesterol level in human cases and rats [7–9].

Expanded application has continued to enhance fermented dairy products containing probiotic bacteria because of their health benefits [10]. Fermented dairy products containing

strains of *Bifidobacteria* and *Lactobacilli* have been developed in several countries around the world to achieve a dietetic therapeutic result that overcomes the symptoms correlated with elevated cholesterol [11]. Lactic acid bacteria can lower blood cholesterol in humans and in experimental animals. Their ability to lower cholesterol is likely due to their lack of binding to bile salts by some bacteria producing the enzyme bile salt hydrolase. The unrelated bile salts are released more in the feces than the associated bile salts [12,13]. The bacteria can secrete bile salt hydrolase that can effectively lower the cholesterol level by stimulating the production of bile salts in conjunction with increasing the manufacture of bile salts of blood cholesterol, practically degrading cholesterol solubility and then decreasing intake in the intestine [12]. Several health benefits have been recorded in the regular uptake of probiotic bacteria, which include lowering cholesterol levels, reducing blood pressure, and facilitating the absorption of salts [14]. It was found that many different bacteria possess such properties, the most important of which are bacteria belonging to the species *Lactobacilli* and *Bifidobacterium*, which is one of several bacteria employed essentially as a dietary supplement [10]. Evidence for the ability of bacteria to lower cholesterol remains unclear, as some strains have shown cholesterol-lowering properties, and others have not, and therefore, research must concentrate on the evaluation of the influence of different strains of bacteria on the biochemical parameters in blood particularly lipid profile [15,16]. High cholesterol as well as high levels of TG and LDL are major risk factors for heart diseases, especially cardiac arrest. Current drugs used to treat heart diseases are not a long-term solution because of their side effects, including digestive disorders, stress, increased activity of liver enzymes, and liver damage, as well as their high costs [17]. Moreover, there is a strong interest in the regulation of blood triglyceride and cholesterol values through the management of any regime that improves lipid profile and specially to include the use of probiotic bacteria in fermented dairy products [18,19]. The present study aimed to assess the health effects of fermented milk manufactured from camel and cow's milk mixture as well as the effect of probiotic bacteria on lowering cholesterol levels and ameliorating the blood lipid profile in hypercholesteremic rats.

## 2. Materials and Methods

### 2.1. Sample Collection

Fresh cow milk and camel milk were obtained from the farm of the College of Agriculture and Veterinary Medicine, Qassim University, Qassim, Saudi Arabia.

### 2.2. Experimental Animals

Forty-eight albino rats (160 ± 10 g) were obtained from the College of Agriculture and Veterinary Medicine, Qassim University, and divided randomly into eight groups (6 rats/group). Animals were housed in a control housing unit at Food Science and Human Nutrition Department, Faculty of Agriculture, Qassim University, and were kept under standard conditions of temperature and humidity (the temperature at 25 °C, 55% humidity and in a 12:12-h light: dark cycle). Rats were fed on a basal diet during the experimental period according to AIN-93 guidelines [20].

### 2.3. High-Fat, High-Cholesterol Diet

High-fat and -cholesterol diet was prepared, containing 67 g standard diet, 31.70 g animal fat, 1% pure cholesterol, and 0.30% bile's acid.

### 2.4. Chemicals

Kits for estimation of serum albumin, total proteins, cholesterol, high-density lipoprotein (HDL), TG, aspartate aminotransferase (AST), alanine aminotransferase (ALT), creatinine, and uric acid kits were obtained from Human, Society for Biochemistry and Diagnostics mbH, Wiesbaden, Germany. LDL was estimated according to the formula of Friedewald et al. [21]:

$$C_{LDL} = C_{Plasma} - C_{HDL} - {}_{TG}/5$$

### 2.5. Starter Cultures and Fermented Milk Manufacture

Two sets of starters were obtained from Christian Hansen (Copenhagen, Denmark). One set of starters is Yo-flex culture (YC-x11), which contains a mixed strain of *Streptococcus thermophilus* and *Lactobacillus bulgaricus* at the ratio of 50:50. The second set is ABT-5 starter culture, which contains a mixed strain of *Streptococcus thermophilus* $TH_4$, *Lactobacillus acidophilus* $LA_5$, and *Bifidobacterium bifidum* $Bb_{12}$ at a ratio of 1:1:1. Different fermented milk were prepared according to the methods described by Tamime and Robinson [22]. Briefly, raw cow and camel milks were heated at 85 °C/10 min, then cooled to 42 °C. The milk was inoculated at the same temperature with 2% activated YC-x11 or ABT-5 starter cultures. The fermentation of yogurt was stopped when the pH value reached 4.6. The gel was stirred and stored at the refrigerator (5 ± 1 °C) until used as shown in Figure 1.

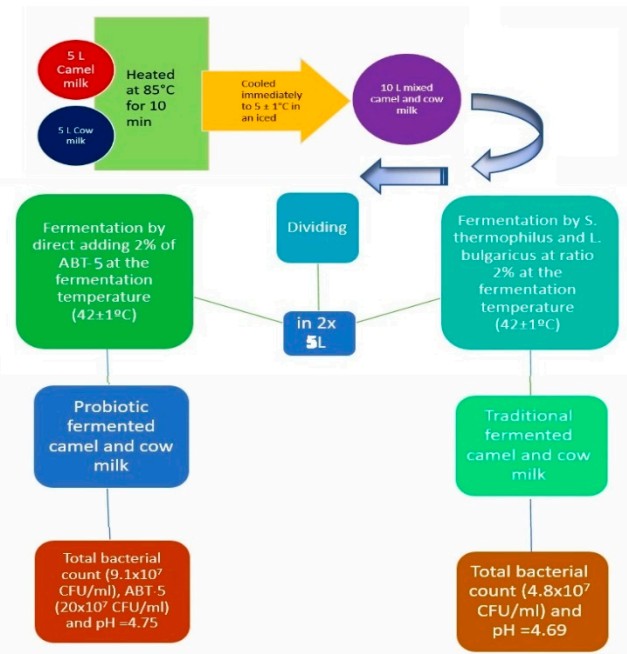

**Figure 1.** Basic process for manufacturing probiotic-fermented mixture of camel and cow milk.

### 2.6. Organoleptic Properties

The evaluation of the sensory properties was conducted in order to quantify the customer favorite for several types of fermented milk as well as the marketing intent. Sensory studies were conducted with students and lecturers of the Food Science and Human Nutrition Department, College of Agriculture and Veterinary Medicine, Qassim University. To perform the sensory analysis, 20 trained and untrained panelists composed the evaluation team, using the scoring described by Tamime and Robinson [22]. Panelists were requested to evaluate the color (10 points), flavor (30 points), acidity (10 points) body and texture (40 points), and overall acceptability (10 points).

### 2.7. Chemical Analysis

The chemical analysis was conducted in different fermented milk samples to determine the pH values and titratable acidity, ash, protein, total solids, lactose, moisture, and fat according to AOAC [23].

### 2.8. Experimental Design

This study was conducted with the approval of the National Committee of Bioethics NCBE at King Abdul-Aziz City for Science and Technology, KACST, KSA, Review Board Number: 10024677(Expiry date: 29 August 2024). Rats were randomly divided into eight groups (*n* = 6) as presented in Figure 2. The first group served as normal control (NC) without further treatment. The other rats switched to a high-fat, high-cholesterol diet

for two weeks and were subdivided into subgroups as follows: group (2) continued on high-cholesterol diet without treatment and was labeled as a positive control (PC). Groups 3 to 8 were treated orally by intestinal tube with 2 mL/day of either fermented cow milk with traditional culture (Co-T), fermented cow milk with ATB-5 (Co-P), fermented camel milk with traditional culture (Ca-T), fermented camel milk with ATB-5 (Ca-P), fermented camel and cow milk mixture with traditional culture (Ca_Co-T), or probiotic-fermented camel and cow's milk mixture with starter culture ATB-5 (Ca_Co-P).

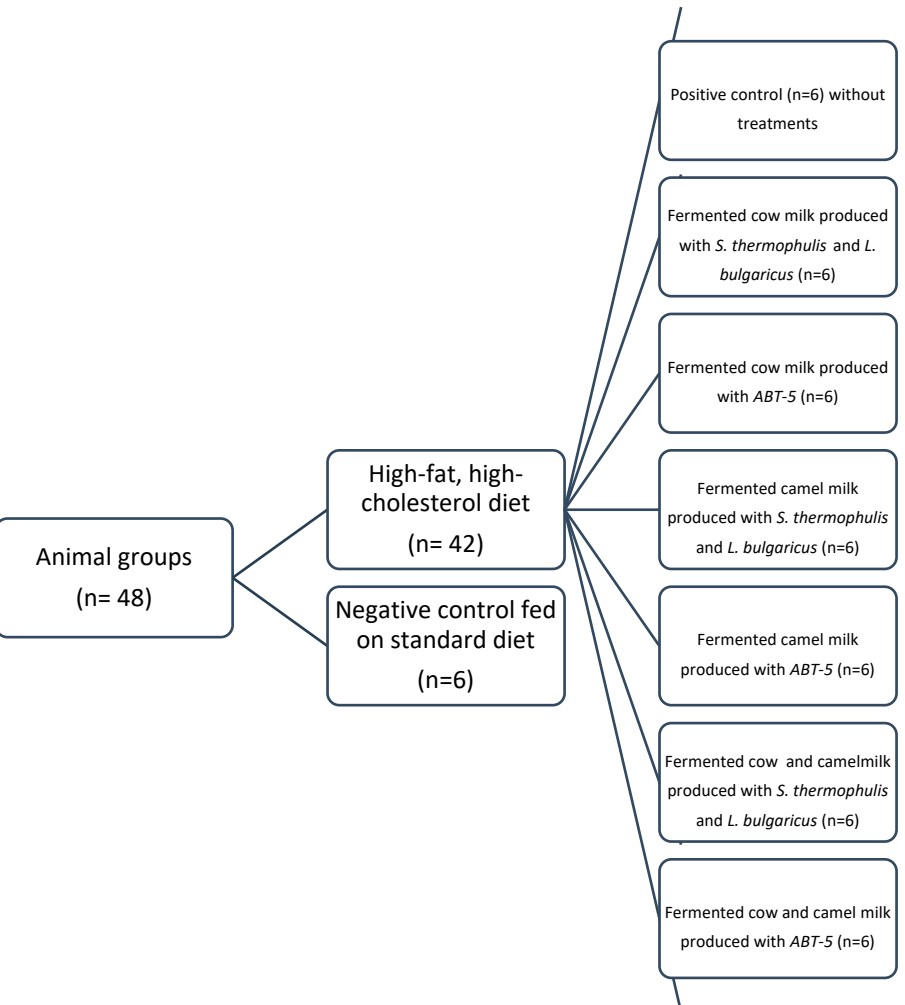

**Figure 2.** Diagram showing experimental design with animal groups.

## 2.9. Biochemical Analysis

Total Cholesterol, HDL, LDL, and triglyceride levels were estimated in serum [24]. Liver enzymes (ALT and AST), serum albumin, and total protein were estimated [25]. Urea and creatinine were determined as indicators of kidney function [26]. All these determinations were performed by commercial kits. All chemicals were purchased from Sigma-Aldrich (St. Louis, MO, USA) Chemical Company. Commercial Kits were purchased from (bio-Merieux Laboratory Reagents and Products, Marcy l'Etoile, France).

## 2.10. Histopathological Examination

Autopsy samples were taken from the liver and kidney of the sacrificed rats and fixed in 10% formalin saline solution for ten hours at least, then washed in tap water for 12 h. Tissue specimens were cleared in xylene and embedded in paraffin. The obtained tissue sections were collected on the glass slides and dyed by hematoxylin and eosin stain for histopathological examination by the light microscope [27].

*2.11. Statistical Analysis*

All experiments as well as related analysis results were performed in triplicate and are presented as means ± standard deviation. A variance analysis (two-way ANOVA) with a significance threshold of $p \leq 0.05$ was used to define the differences between groups. Statistica 12.5 software was used to conduct the analysis (Stat Soft Inc., Tulsa, OK, USA).

## 3. Results and Discussion

*3.1. Chemical Composition of Various Fermented Milk*

An initial experiment was carried out to prepare a mixture of fermented camel and cow's milk in different proportions (25:75, 50:50, and 75:25% *v/v*), and then, fermented milk was produced by using ABT-5 probiotic strains and assessed the organoleptic characteristics and the level of consumers' accessibility to it. It was found that the best mixing ratios for camel and cow's milk was (50: 50% *v/v*), which was followed in the biological experiment to estimate the effect of probiotic bacteria as well as the effect of camel milk on the lipid profile in rats fed on a high-fat diet (results not showed). Results summarized in Table 1 showed that the fat content in probiotic-fermented milk increased slightly in comparison with the fermented milk produced with the traditional starter culture. The amount of lactic acid produced increased with a concomitant drop in pH with an increase in fermentation time. The pH of fermented camel milk with ABT-5 was greater than that prepared by using traditional starter cultures. However, the combination of *L. bulgaricus* and *S. thermophilus* (1:1) resulted in a lower pH and higher acidity than probiotic starter cultures. In addition, fermented milk manufactured with the traditional starter was higher in acidity than probiotic-fermented milk. These results are in accord with the results reported by Ibrahem [28], which reported an increase in acidity up to 50% when storing fermented camel milk for 24 h after coagulation. This increase in acidity may be due to the difference in the metabolic activities of microbial species used as starters in the fermented milk manufacturing. The decrease in total protein in fermented milk samples occurred as a result of protein hydrolysis by the starter culture. These results agree with the results obtained by Galeboe [29]. Protein degradation in all treatments increased due to the limited hydrolysis of milk proteins by lactic acid bacteria.

**Table 1.** Chemical composition of different fermented milks.

| Attribute | Fermented Milk Types | | | | | |
|---|---|---|---|---|---|---|
| | **Co-T** | **Co-P** | **Ca-T** | **Ca-P** | **Ca_Co-T** | **Ca_Co-P** |
| Moisture (%) | 85.66 ± 1.2 [b] | 85.00 ± 1.2 [b] | 87.48 ± 1.3 [a] | 87.37 ± 1.3 [a] | 86.62 ± 1.3 [a,b] | 86.46 ± 1.4 [a,b] |
| Fat (%) | 4.80 ± 0.33 [a] | 4.60 ± 0.33 [a] | 3.3 ± 00.16 [c] | 3.50 ± 0.20 [c] | 4.35 ± 0.12 [b] | 4.50 ± 0.33 [b] |
| Protein (%) | 4.83 ± 0.21 [a] | 4.63 ± 0.21 [a] | 4.39 ± 0.08 [b] | 4.40 ± 0.10 [b] | 4.47 ± 0.14 [a,b] | 4.49 ± 0.24 [a,b] |
| Ash (%) | 0.78 ± 0.07 [b,c] | 0.75 ± 0.07 [c] | 0.86 ± 0.10 [a] | 0.84 ± 0.10 [a] | 0.80 ± 0.08 [b] | 0.82 ± 0.09 [a,b] |
| Lactose (%) | 3.93 ± 0.17 [a,b] | 3.83 ± 0.17 [b] | 3.97 ± 0.22 [a] | 3.89 ± 0.32 [b] | 3.76 ± 0.16 [c] | 3.73 ± 0.14 [c] |
| pH | 4.83 ± 0.10 [a] | 4.73 ± 0.10 [b,c] | 4.71 ± 0.13 [b,c] | 4.76 ± 0.12 [b] | 4.72 ± 0.12 [b,c] | 4.69 ± 0.09 [c] |
| Acidity (%) | 0.78 ± 0.11 [a] | 0.75 ± 0.11 [a,b] | 0.71 ± 0.09 [b] | 0.74 ± 0.11 [a,b] | 0.75 ± 0.11 [a,b] | 0.70 ± 0.11 [b] |

Data are presented as the mean ± SD (n = 3/group). Different superscript letters (a to b) within the same row show significant differences among the groups ($p \leq 0.05$). NC, normal rats fed on basal diet; PC, rats fed on high-fat, high-cholesterol diet without treatments; Co-T, fermented cow milk with traditional starter culture; Co-P, fermented cow milk with ABT-5 strains; Ca-T, fermented camel milk with traditional starter culture; CAP, fermented camel milk with ABT-5; Ca_Co-T, fermented camel and cow milk with traditional starter culture; Ca_Co-P, fermented camel and cow milk with ABT-5.

### 3.2. Sensory Evaluation of Different Fermented Milks

Five samples of camel and cow's milk fermented for 6 h at 42 ± 1 °C by selected starter cultures were prepared and sensory evaluated by 10 untrained panelists for color, body and texture, taste and odor, acidity, and overall acceptability. The mean values of sensory evaluation scores are summarized in Table 2. The results showed that there were no significant differences ($p \leq 0.05$) in the color of the fermented products. In general, the panelists gave lower sensory scores for body and texture for both traditional and probiotic-fermented camel milk. However, the consistency of all fermented camel milk products was watery and showed a fragile and heterogeneous structure. The overall acceptability scores of the sensory evaluation revealed that the cow-fermented milk by yogurt starter culture was the most accepted, followed by the cow and camel fermented milk, while probiotic-fermented camel milk was the least. Low sensory assessment of fermented camel milk can be due to high-saturated fatty acid contents, salts, weak lipolysis, and poor protein structure [5,30]. The principal factor responsible for improvements in sensory properties of fermented milks was the development of traditional starter culture and probiotic bacteria during fermentation or storage period. The results of this study were consistent with the results obtained by Boukria [31].

**Table 2.** Sensory evaluation of different fermented milks.

| Attribute | Fermented Milk Types | | | | | |
|---|---|---|---|---|---|---|
| | **Co-T** | **Co-P** | **Ca-T** | **Ca-P** | **Ca_Co-T** | **Ca_Co-P** |
| Flavor (30) | 28.91 ± 0.3 [a] | 28.61 ± 0.3 [a] | 27 ± 0.56 [b,c] | 26.8 ± 0.24 [c] | 28.13 ± 0.78 [b] | 28.5 ± 0.63 [a] |
| Color (10) | 9.2 ± 0.12 [a] | 9.2 ± 0.12 [a] | 9.1 ± 0.17 [a,b] | 8.85 ± 0.19 [c] | 9.0 ± 0.2 [b] | 9.1 ± 0.24 [a,b] |
| Body and Texture (40) | 36.9 ± 0.4 [a] | 36.1 ± 0.4 [b] | 34.17 ± 0.82 [c] | 33.52 ± 0.27 [d] | 36.5 ± 0.24 [a,b] | 36.0 ± 0.33 [b] |
| Acidity (10) | 9.02 ± 0.2 [a] | 8.92 ± 0.2 [a,b] | 8.62 ± 0.16 [b,c] | 8.54 ± 0.1 [c] | 8.58 ± 0.18 [c] | 8.81 ± 0.16 [b] |
| Overall acceptability (10) | 9.11 ± 0.13 [a] | 9.01 ± 0.13 [a,b] | 8.64 ± 0.17 [c] | 8.65 ± 0.12 [c] | 8.9 ± 0.21 [b] | 8.92 ± 0.23 [b] |
| Total (100) | 93.14 ± 0.2 [a] | 91.84 ± 0.2 [b] | 87.53 ± 0.38 [c] | 86.36 ± 0.24 [c] | 91.11 ± 0.4 [b,c] | 91.33 ± 0.33 [b] |

Data are presented as the mean ± SD (n = 3/group). Different superscript letters (a to d) within the same row show significant differences among the groups ($p \leq 0.05$). NC, normal rats fed on basal diet; PC, rats fed on high-fat, high-cholesterol diet without treatments; Co-T, fermented cow milk with traditional starter culture; Co-P, fermented cow milk with ABT-5 strains; Ca-T, fermented camel milk with traditional starter culture; CAP, fermented camel milk with ABT-5; Ca_Co-T, fermented camel and cow milk with traditional starter culture; Ca_Co-P, fermented camel and cow milk with ABT-5.

### 3.3. Effect of Fermented Milk on the Body Weight Gain in Rats

Table 3 shows the effect of oral dosage of various fermented milk (2 mL/day) on body weight of rats fed on a high-fat, high-cholesterol diet. It was found that fermented milk contributed to a decrease in fat levels in rat blood coupled with a relative weight reduction. The results indicated significant variations between rat groups fed on different fermented milks: 21%, 20.3%, 21.7%, 17.3%, 23.7%, and 24.75%, respectively, compared to the positive control group (32.7%). Our findings are in agreement with those previously reported [32–34].

**Table 3.** Effect of different fermented milks on some growth parameters in hypercholesterolemic rats.

| Groups | Initial Body Weight (g) | Final Body Weight (g) | Body Weight Gain (%) |
|---|---|---|---|
| NC | 265.4 ± 11.7 [c] | 319.3 ± 12.7 [c] | 20.38 ± 0.2 [b,c] |
| PC | 287.3 ± 11.9 [a,b] | 381.3 ± 13.7 [a] | 32.71 ± 1.62 [a] |
| Ca-T | 291.33 ± 10 [a,b] | 352.3 ± 11.9 [b] | 20.97 ± 0.5 [b,c] |
| Ca-P | 290.2 ± 12.8 [a] | 349.3 ± 12.6 [b] | 20.34 ± 0.3 [b,c] |
| Co-T | 285.2 ± 10.3 [a,b] | 347.3 ± 7.4 [b] | 21.71 ± 0.8 [b,c] |
| Co-P | 295.3 ± 9.3 [a] | 346.5 ± 6.4 [b,c] | 17.29 ± 0.8 [c] |
| Ca_Co-T | 279.2 ± 11.2 [b] | 345.7 ± 10.8 [b,c] | 23.66 ± 0.7 [b] |
| Ca_Co-P | 290.7 ± 10.2 [a] | 361.9 ± 10.5 [a,b] | 24.75 ± 0.9 [b] |

Data are presented as the mean ± SD (n = 3/group). Different superscript letters (a to c) within the same row show significant differences among the groups ($p \leq 0.05$). NC, normal rats fed on basal diet; PC, rats fed on high-fat, high-cholesterol diet without treatments; Co-T, fermented cow milk with traditional starter culture; Co-P, fermented cow milk with ABT-5 strains; Ca-T, fermented camel milk with traditional starter culture; CAP, fermented camel milk with ABT-5; Ca_Co-T, fermented camel and cow milk with traditional starter culture; Ca_Co-P, fermented camel and cow milk with ABT-5.

*3.4. Effect of Fermented Milks on Blood Lipid Profile*

The positive control group showed a significant ($p \leq 0.05$) elevated serum total cholesterol, TG, and LDL, but HDL was significantly ($p \leq 0.05$) lower throughout the experimental period. Table 4 shows that total cholesterol levels in the positive control group (195 mg/dL) increased significantly compared with the negative control group (74 mg/dL). The level of cholesterol was 113, 112.3, 103, 97, 95.3, and 89.5 mg/dL, respectively, in rat doses with different fermented milks (Co-T, Co-P, Ca-T, Ca-P, Ca_Co-T, and Ca_Co-P). Abdelgawad [35] and El-Zahar [19] found that yogurt made by using *Bif. longum* and *L. acidophilus* led to a decrease in total cholesterol levels in high-cholesterol diet rats. The level of triglycerides increased significantly because of feeding on high-fat, high-cholesterol diets (159 mg/dL) compared to the negative control group (43.6 mg/dL). Rats treated with probiotic-fermented camel and cow milk showed the lowest level of triglycerides (75.4 mg/dL) compared to other groups (Ca_Co-T, Ca-P, Ca-T, Co-P, and Co-T): 83.2, 84.9, 86.5, 100.6, and 102 mg/dL, respectively. These effects associated with lowering triglycerides may be associated with decreasing body fat and accelerating fat metabolism [36]. Probiotic bacteria metabolize indigestible lactose to produce short-chain fatty acids in the gut, which causes a decrease in the systemic blood lipids by inhibiting the hepatic cholesterol synthesis and redistributing cholesterol from plasma to the liver [37]. Results in Table 4 show that feeding with fermented camel and cow milk produced from ATB-3 starter culture increases the level of HDL (38.3 mg/dL) compared to the PC group (30.5 mg/dL). This study is consistent with previous studies [38–40]. They found that probiotic bacteria strains have attracted attention as potential cholesterol-lowering milk additives. Feeding on the probiotic-fermented camel and cow milk significantly reduced LDL level (37.6 mg/dL) as compared to traditional fermented camel and cow mixture (45.7 mg/dL). Generally, feeding on different fermented milks was found to decrease LDL levels in hypercholesterolemic rats as compared to the positive control group (133.9 mg/dL). The levels of LDL decreased from 64.9, 63.8, 47.8, 44.9, 45.7, and 37.6 mg/dL in Co-T, Co-P, Ca-T, Ca-P, Ca_Co-T, and Ca_Co-P groups, respectively.

Fermented camel milk and fermented cow milk containing bifidobacterium were very effective in lowering the level of plasma and liver lipids in rats. These hypocholesterolemic effects of fermented cow milk containing bifidobacterium, which have been demonstrated in the rats in the present study, could make an effective and economic contribution in treating hypercholesterolemia if these effects could be confirmed in human volunteers [6]. Fermented camel may decrease cholesterol and triglyceride levels by many proposed mechanisms: (1) inhibition of liver cholesterol synthesis and distribution of cholesterol

from the blood circulation to the liver; (2) intestinal bacteria that may inhibit the absorption of cholesterol; and (3) milk fermented by lactic acid strains may inhibit cholesterol synthesis enzymes and decrease cholesterol level [8,9,16]. Moreover, bioactive peptides derived from camel milk proteins may reduce cholesterol through the interaction between bioactive peptides and cholesterol and the presence of orotic acid in camel milk, which is thought to be responsible for lowering cholesterol level in rats [6,13].

**Table 4.** Effect of different fermented milks on blood lipid profile in hypercholesterolemic rats.

| Groups | TG mg/dL | TC mg/dL | HDL mg/dL | LDL mg/dL | VLDL mg/dL |
|---|---|---|---|---|---|
| NC | 43.6 ± 1.1 [e] | 77.6 ± 1.7 [f] | 41.7 ± 0.8 [a] | 31.2 ± 1.3 [e] | 12.2 ± 0.1 [e] |
| PC | 159.2 ± 2.1 [a] | 195.4 ± 1.2 [a] | 30.5 ± 0.4 [d] | 133.9 ± 1.7 [a] | 34.3 ± 0.5 [a] |
| Ca-T | 86.5 ± 1.5 [c] | 102.9 ± 2.2 [c] | 33.4 ± 1.5 [c] | 47.9 ± 1.6 [c] | 19.5 ± 0.3 [c] |
| Ca-P | 84.9 ± 1.1 [c] | 97.1 ± 1.6 [c,d] | 36.2 ± 1.7 [b] | 44.8 ± 1.6 [c] | 19.2 ± 0.4 [c] |
| Co-T | 101.9 ± 1.3 [b] | 113.3 ± 2.6 [b] | 33.7 ± 1.3 [c] | 64.9 ± 1.5 [b] | 23.6 ± 0.5 [b] |
| Co-P | 100.6 ± 1.2 [b] | 112.3 ± 2.4 [b] | 32.6 ± 1.5 [c] | 63.8 ± 1.5 [b] | 22.8 ± 0.5 [b] |
| Ca_Co-T | 83.2 ± 0.8 [c] | 95.3 ± 2.1 [d] | 35.9 ± 0.6 [b] | 45.7 ± 2.2 [c] | 17.3 ± 0.5 [c,d] |
| Ca_Co-P | 75.4 ± 0.6 [d] | 89.5 ± 1.2 [e] | 38.3 ± 1.2 [a,b] | 37.6 ± 2.5 [d] | 16.3 ± 0.7 [d] |

Data are presented as the mean ± SD (n = 3/group). Different superscript letters (a to f) within the same row show significant differences among the groups ($p \leq 0.05$). NC, normal rats fed on basal diet; PC, rats fed on high-fat, high-cholesterol diet without treatments; Co-T, fermented cow milk with traditional starter culture; Co-P, fermented cow milk with ABT-5 strains; Ca-T, fermented camel milk with traditional starter culture; CAP, fermented camel milk with ABT-5; Ca_Co-T, fermented camel and cow milk with traditional starter culture; Ca_Co-P, fermented camel and cow milk with ABT-5.

### 3.5. Effect of Different Fermented Milks on Liver Functions

Hypercholesterolemia has been known to disturb the oxidant–pro-oxidant balance and reduce the efficacy of the antioxidant protection system, leading to tissue damage, frequently correlated with the evolution and development of atherogenesis [41]. Lipid peroxidation is an oxidative modification of polyunsaturated fatty acids in the cell layers that produce several degeneration products. Certain products cause great disturbance to cell elements and may prevent cell replication and cell endurance. The activities of AST and ALT were assayed in serum samples as the liver function biomarkers. The results in Table 5 indicated that feeding on high-fat, high-cholesterol diet during the experimental period resulted in damage to hepatocytes, leading to high levels of ALT and AST in the positive control group (84.2 and 119.3 U/L) compared to the negative control group (37.4 and 47.7 U/L), respectively. Generally, feeding on different fermented milks were found to decrease AST and ALT levels in hypercholesterolemic rats as compared to the positive control group, where the levels of AST decreased from 74.7, 73.3, 99.3, 95.2, 82.3, and 74.5 mg/dL in Co-T, Co-P, Ca-T, Ca-P, Ca_Co-T, and Ca_Co-P groups, respectively. These results suggest that those hepatic biomarkers were raised in the serum due to the release of the enzymes from damaged livers. Nevertheless, a significant reduction ($p \leq 0.05$) was observed in rats fed with different kinds of fermented milk compared with the positive control group. This decrease in aminotransferase enzymes and the recovery of hepatocytes for some of their vital functions can be attributed to the high content antioxidants in camel milk, which preserve the plasma membrane of the hepatocytes and protect it from rupture and exit of the cytosol loaded with these enzymes [42]. Camel milk is rich in the antioxidant vitamin C. Vitamin C concentration in camel milk is 3.5 times greater (51 mg/L) than in cow's milk (15 mg/L), and it plays a vital role in the protection of hepatocytes from lipid peroxidation. The protective effect may also be attributed to the role of starter cultures used in the fermented dairy industry [19,38,39]. These results are consistent with the results obtained previously [20,36], which reported a significant increase in cardiac enzyme activities due to increased ALT synthesis in the liver due to high cholesterol. The

improvement in liver function may result from feeding groups of rats either conventional fermented milk or probiotic, and this was confirmed by the histological examination of the liver. At the end of the experiment, hepatic tissue appeared almost natural, without congestion or cellular infiltration. The reason for enhanced serum enzymes in high-fat diet-induced liver injury by camel milk may occur due to the prevention of the leakage of intracellular enzymes by membrane-stabilizing action. Numerous researches should provide substantial support for evidencing the protecting effects of camel milk on liver injury [43]. Total protein and albumin levels shown in Table 5 revealed a significant drop in the positive control group compared to other groups. This may be attributed to the damage in the liver caused by hypercholesterolemia, which in turn leads to a decrease in the ability of the liver to synthesize protein, or may be due to high-calorie diets, which impair the absorption of protein and other nutrients [44]. It is clear from Figure 3 that there are significant differences due to feeding on different fermented milks in either albumin or total protein, where the albumin values ranged between 3.5–4.2 g/dL, and total protein ranged from 6.5 to 7.75 g/dL respectively. Albumin levels in treated rats with different fermented milks (Co-P, Co-T, Ca-T, Ca_Co-T, Ca-P, and Ca_Co-P) were significantly increased by 3.5, 3.6, 3.9, 3.9, 4.1, and 4.2 mg/dL, respectively. The increase in total protein and albumin levels may be due to the ability of lactic acid and probiotics to control changes in blood lipid profile, thereby reducing damage to liver tissue caused by increased blood lipids and cholesterol. The results obtained agreed with those obtained by other researchers [45,46] that confirmed feeding on yogurt, which was produced by using probiotic bacteria, significantly improved total protein and albumin levels and indicated that the enhancement effect was due to the use of probiotics.

**Table 5.** Effect of different fermented milks on albumin, protein, serum enzyme activities of liver, and kidney markers in hypercholesterolemic rats.

| Groups | AST (U/L) | ALT (U/L) | Albumin (g/dL) | Total Protein (g/dL) | Creatinine (mg/dL) | Urea (mg/dL) |
|---|---|---|---|---|---|---|
| NC | 47.67 ± 1.9 [e] | 37.4 ± 0.6 [e] | 3.8 ± 0.1 [b,c] | 7.28 ± 0.14 [b] | 0. 92 ± 0.1 [d] | 14. 4 ± 0.2 [f] |
| PC | 119.3 ± 1.6 [a] | 84.2 ± 1.0 [a] | 2.9 ± 0.03 [d] | 5.48 ± 0.14 [f] | 1.6 ± 0.1 [a] | 37.89 ± 0.2 [a] |
| Ca-T | 74.7 ± 1.0 [d] | 46.6 ± 0.7 [c] | 3.9 ± 0.1 [b] | 7.31 ± 0.1 [b] | 1.2 ± 0.1 [b,c] | 25.1 ± 0.4 [c] |
| Ca-P | 73.3 ± 0.6 [d] | 45.1 ± 1.6 [c] | 4.1 ± 0.1 [a] | 7.5 ± 0.13 [a] | 1.1 ± 0.07 [c] | 22.2 ± 0.2 [d] |
| Co-T | 99.3 ± 0.6 [b] | 51.7 ± 0.4 [b] | 3.6 ± 0.02 [c] | 6.94 ± 0.06 [c] | 1.3 ± 0.1 [b] | 27.57 ± 0.4 [b] |
| Co-P | 95.2 ± 0.5 [b] | 52.4 ± 0.4 [b] | 3. 5 ± 0.02 [c] | 6.5 ± 0.06 [e] | 1.3 ± 0.1 [b] | 26.23 ± 0.3 [b] |
| Ca_Co-T | 82.3 ± 0.63 [c] | 43.1 ± 0.7 [c,d] | 3.9 ± 0.1 [b] | 7.10 ± 0.1 [d,e] | 1.13 ± 0.1 [c] | 23 ± 0.2 [d] |
| Ca_Co-P | 74.45 ± 2.8 [d] | 41.3 ± 0.63 [d] | 4.2 ± 0.1 [a] | 7.76 ± 0.1 [a] | 1.01 ± 0.05 [c,d] | 19.5 ± 0.3 [e] |

Data are presented as the mean ± SD (n = 3/group). Different superscript letters (a to f) within the same row show significant differences among the groups ($p \leq 0.05$). NC, normal rats fed on basal diet; PC, rats fed on high-fat, high-cholesterol diet without treatments; Co-T, fermented cow milk with traditional starter culture; Co-P, fermented cow milk with ABT-5 strains; Ca-T, fermented camel milk with traditional starter culture; CAP, fermented camel milk with ABT-5; Ca_Co-T, fermented camel and cow milk with traditional starter culture; Ca_Co-P, fermented camel and cow milk with ABT-5.

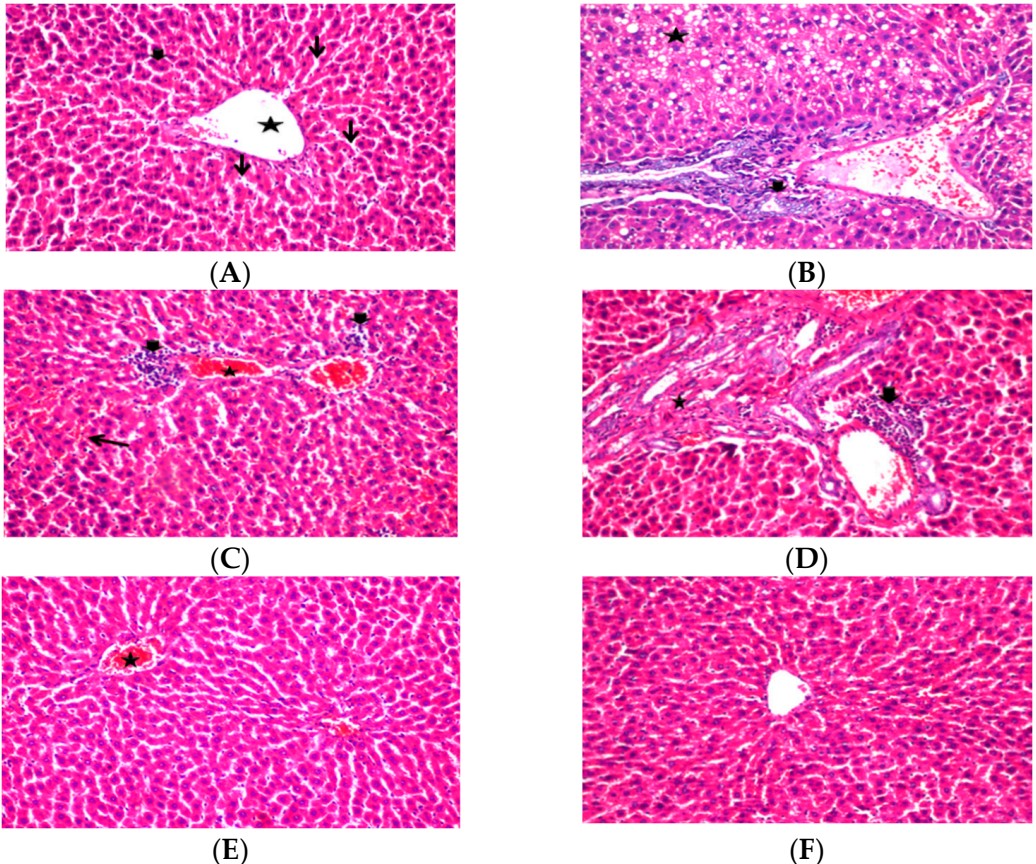

**Figure 3.** Histopathological examination of liver tissue (H&E x200) showing damage and arrangement of hepatocytes and fat vacuoles; (**A**) tissue from the NC group showing a normal liver with no lipid deposits; (**B**) liver tissue from the PC group showing the presence of steatosis; (**C**) tissue from the Co-T group showing congested blood vessels and sinusoids, Kupffer cells hyperplasia, and lymphocytic infiltrations; (**D**) tissue from the Ca-T group showing peri-artery inflammatory cells aggregations mainly lymphocytes; (**E**) cells from the Ca_Co-T group showing nearly normal hepatic lobules with mild congested central vein; and (**F**) tissue from the Ca_Co-P group showing apparently normal hepatic lobe.

### 3.6. Effect of Fermented Milks on Renal Function in Hypercholesterolemic Rats

Results revealed a significant ($p \leq 0.05$) rise in creatinine and urea in rats fed on a high-cholesterol diet, and when comparing between groups, it was noticed that there was a decrease in the levels of creatinine and urea in rats fed on the probiotic-fermented camel and cow milk (1.01 and 19. 5 mg/dL). Urea and creatinine levels increased significantly ($p < 0.05$) in rats treated with high-fat, high-cholesterol diet (37.9 and 1.6 mg/dL) compared with the negative control one (14.4 and 0.92 mg/dL), respectively (Table 5). Generally, ingestion of fermented milk containing probiotics was more efficient in decreasing creatinine and urea concentrations than feeding on traditional fermented milks. These results were confirmed by histological examination of the kidney, which showed some glomeruli crowded with some renal pellets. This is consistent with results obtained by previous researchers showing that a significant increase in creatinine concentrations in blood serum in rats fed a high-cholesterol diet, increasing the risk of developing renal injury [45,47]. Urea and uric acid are the principal waste products of protein catabolism. They are synthesized in the liver from the ammonia, which produced as a result of amino acids de-amination [48,49], while creatinine is the major waste product of creatine metabolism by muscle. In the kidney, creatinine is distilled by the glomerulus and excreted by the tubules, and only free creatinine appears in the blood serum. The decrease in creatinine and urea may be due to the ability

of lactic acid and probiotic bacteria to control the alteration of lipid profile, leading to improvement in the renal damage due to on the intake of high-fat, high-cholesterol diet.

### 3.7. Histological Changes in Experimental Rats

The liver in control rats showed that hepatocytes, portal triads, and vasculature appeared normal (Figure 3A). The hepatic sinusoids were certainly evident, and the hepatic cord was usually provided. Nevertheless, diffuse lipid alterations were observed in rats fed on the high-fat, high-cholesterol diet (Figure 3B). There was a marked improvement in hepatic tissue with very few hydrolyzed hepatocytes in liver tissue in rats fed on high-fat, high-cholesterol diet and then dosed with fermented cow milk as shown (Figure 3C). The liver damage in different groups (Figure 3E,F, respectively) was decreased, the number of fatty liver cells was reduced, and the number of lipid droplets in the cytoplasm was also lowered. Liver tissue in rats in group Ca_Co-T showed significant improvement in liver cells, and hepatic tissue returned to normal without congestion or any inflammatory cellular infiltration (Figure 3E). Ca_Co-P group demonstrated a complete regeneration structure of liver tissue by gaining its natural form, as it was a guide for managing the damage done by high-fat diet (Figure 3F). These results agree with the results previously obtained [19,50], which reported the administration of the transcription of lipid metabolism-related genes affected by Bifidobacterium-fermented milk was more powerful than that by the mixed probiotic-fermented milk. The kidney parenchyma in the negative control group was arranged not to show any morphological variations (Figure 4A). Kidney segments in the high-fat, high-cholesterol diet group pointed to several alterations in the cortex and external medulla; in some glomeruli, a decrease of the filtration chamber and glomerular tuft adhering to Bowman's capsule were obvious. Furthermore, hyaline material and hemorrhagic casts were visible (Figure 4B). All other groups showed improvement in kidney tissues compared to the positive control group, where there were no signs of infection, congestion, or signs of inflammation (Figure 4C–F, respectively). The histological examination showed the influence of lactic acid bacteria and *Bif. Bifidum* and *L. acidophilus* in fermented milk and some elements of camel milk in repairing these damages (Figure 4E,F). Moreover, the impact of high-fat, high-cholesterol diet on the renal tissue caused congestion in some renal blood vessels, including renal glomeruli in the renal pellets, making the tissue satisfactory. However, lactic acid bacteria and probiotic bacteria played an important role in the gradual restoration of hepatocellular tissue to a different degree from one treatment to another. The impact of probiotic bacteria in reducing the pathological effects of hypercholesterolemic rats was more effective than traditional starter cultures. In this regard, the reduction in TG in the blood serum in hypercholesterolemic rats may be associated with the production of uric acid by the initiator used, which was found to lower the level of triglycerides in the blood [36,51]. The liver is inclined to xenobiotic-induced harm because of its basic function in xenobiotic metabolism and its portal situation inside the circulatory system. No significant difference was discovered in tissue sections of the liver of treated rats maintained with probiotic-fermented mixture milk and negative control groups. All the segments were natural, without any inflammatory lesions. Different kinds of probiotic-fermented milk offered to the rats produced no unfavorable results in the histology of the liver.

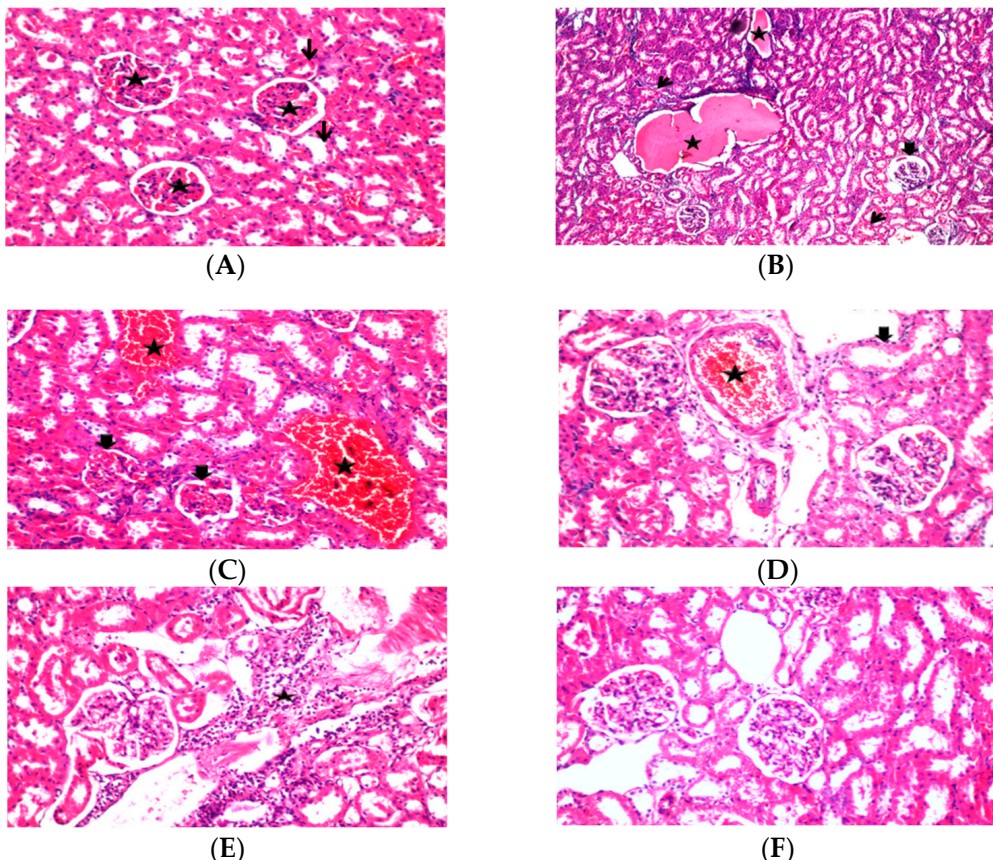

**Figure 4.** Hematoxylin-eosin (H&E x300) stained sections of rat kidneys; (**A**) normolipidemic controls, NC; (**B**) hyperlipidemic controls, PC; (**C**) rats treated with cow fermented milk using traditional starter culture; (**D**) rats treated with camel fermented milk using traditional starter culture; (**E**) rats treated with fermented camel and cow's milk using traditional starter culture; and (**F**) kidney sections of rats treated with fermented camel and cow's milk using probiotic culture.

## 4. Conclusions

In conclusion, the present study indicated that camel milk mixed with cow milk and camel milk alone influenced the beneficial effect much more than cow's milk, and probiotic starter culture had a significantly stronger effect than traditional yogurt culture. Camel milk fermented with ABT-5 significantly reduced risk of dyslipidemia associated with the metabolic syndrome in hypercholesterolemic rats, lowering body-weight gain and serum triglycerides and LDL while boosting serum HDL. Biomarkers of kidney function (urea, creatinine) also increased dramatically, but total protein content in serum and serum albumin dropped moderately. Camel milk fermented with probiotic bacteria significantly ($p \leq 0.05$) improved the lipid levels and intuitive lipid buildup in hyperlipidemic rats. These metabolic improvements were accompanied by a reduction in tissue damage (liver, kidney). Results verified that feeding on the fermented milk with ATB-5 strains enhances the body's production of inhibitors that lower cholesterol levels and improve liver and kidney functions. Camel milk is superior to cow milk in this respect. This study encourages the consumption of fermented camel milk, as obesity and dyslipidemia are widely spread in the Arab countries.

**Author Contributions:** Conceptualization, K.M.E.-Z.; Data curation, H.M.M.; Formal analysis, Y.M.A.; Investigation, K.M.E.-Z. and H.M.M.; Methodology, Y.M.A.; Project administration, K.M.E.-Z.; Validation, Y.M.A.; Visualization, H.M.M. All authors have read and agreed to the published version of the manuscript.

**Funding:** Researchers would like to thank the Deanship of Scientific Research, Qassim University for funding the publication of this project.

**Institutional Review Board Statement:** This study was conducted with the approval of the National Committee of Bioethics NCBE at King Abdul-Aziz City for Science and Technology, KACST, Kingdom of Saudi Arabia, Review Board Number: 10024677.

**Informed Consent Statement:** Not applicable.

**Data Availability Statement:** The authors declare the availability of data and material; they also declare the data are transparent for this MS.

**Conflicts of Interest:** The authors declare no conflict of interest in line with the required format. The authors are not part of any associations or commercial relationships that might represent conflicts of interest in the writing of this study.

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
