# Peer review of "Beneficial Effects of Fermented Camel and Cow’s Milk in Lipid Profile, Liver, and Renal Function in Hypercholesterolemic Rats"

_fermentation, doi:10.3390/fermentation8040171_

Round 1

Reviewer 1 Report

The manuscript is well written. 

Author Response

Dear Reviewer 

It is my pleasure to thank you for your useful comments and all corrections pointed out in the text were accepted and done

Reviewer 2 Report

Very interesting topic. Good experimental design.  More references should be added to support results and discuss the findings. More references are needed on the mechanisms used by fermented milk (cow and camel) to lower cholesterol. 

Some of the diagrams are too big. More details should be included on the methods used for chemical analysis.

Line 341- the discussion of the effect of fermented milk on lowering creatinine. This is a very relevant finding, but there is not enough references to support the conclusions.  Discussion could be improved. 

Author Response

Dear Reviewer 

Thank you very much for your useful comments which was taken into consideration.

1- Minor spell check was made and corrected 

2- The introduction was improved by inclusion of relevant references to support the background of the study.

3-The conclusions were improved and supported by results.

4-More references were added to support the results and discussion of the findings.

5-More references were added to explain the mechanism used by fermented milk (cow+ camel) to lower cholesterol.

6-More details were added to the methods used for chemical analysis  .

7.The effect of fermented milk in lowering creatinine was discussed and new references were added to support the conclusions.

This manuscript is a resubmission of an earlier submission. The following is a list of the peer review reports and author responses from that submission.